# Dust-Acoustic Nonlinear Waves in a Nanoparticle Fraction of Ultracold (2K) Multicomponent Dusty Plasma

**DOI:** 10.3390/molecules27010227

**Published:** 2021-12-30

**Authors:** Fedor M. Trukhachev, Roman E. Boltnev, Mikhail M. Vasiliev, Oleg F. Petrov

**Affiliations:** 1Joint Institute for High Temperatures of the Russian Academy of Sciences, 125412 Moscow, Russia; boltnev@gmail.com (R.E.B.); mixxy@mail.ru (M.M.V.); ofpetrov@ihed.ras.ru (O.F.P.); 2Belarusian-Russian University, 212000 Mogilev, Belarus; 3Moscow Institute of Physics and Technology, 141701 Dolgoprudny, Russia; 4Chernogolovka Branch of the N.N. Semenov Federal Research Center for Chemical Physics, Russian Academy of Sciences, 142432 Chernogolovka, Russia

**Keywords:** ultracold dusty plasma, nonlinear dusty-acoustic wave, Debye radius

## Abstract

The nonlinear dust-acoustic instability in the condensed submicron fraction of dust particles in the low-pressure glow discharge at ultra-low temperatures is experimentally and theoretically investigated. The main discharge parameters are estimated on the basisof the dust-acoustic wave analysis. In particular, the temperature and density of ions, as well as the Debye radius, are determined. It is shown that the ion temperature exceeds the temperature of the neutral gas. The drift characteristics of all plasma fractions are estimated. The reasons for the instability excitation are considered.

## 1. Introduction

Dusty plasma is a plasma containing fractions of charged microparticles along with electrons and ions. Such plasmas are common in space and have many technological applications [1]. The main methods of dust fraction formation are injection of dust particles in plasma, chemical reactions in plasma, and ion sputtering of various materials in plasma [1]. In rare and poorly studied cases, several mechanisms are involved simultaneously [2]. The presence of low-frequency wave modes is one of the features of dusty plasma. For example, the frequencies of well-understood dust-acoustic waves are in the range of 0.1–100 Hz.

In recent years, studies of cryogenic dusty plasma have become relevant, which is due to various fundamental and technological aspects [3,4,5,6,7,8,9,10]. Notably, the study of ultra-low-temperature discharges is an independent physical problem [11,12,13]. Among the open questions, one can single out the gas-to-ion temperature ratio, the magnitude of the screening scale, etc. For example, dusty plasma in a glow discharge at liquid nitrogen and helium temperatures was investigated in [4]. It was assumed that the temperatures of ions and neutral gas are equal. In this case, the ion Debye radius is of the order of one micron at a gas temperature of 4.2 K, pressure *p* = 4 Pa, and the ratio of the electric field to the density of a neutral gas *E*/*N*~10 Td. The same results are presented in a study by [6]. On the other hand, in [9], by studying the dust interparticle distance, it was found that the screening radius exceeds the Debye ionic radius by an order of magnitude assuming that the ionic and atomic temperatures are equal. Possible explanations for this phenomenon can be a nonlinear screening of a particle charge [14,15], overheating of ions in the discharge plasma [16,17,18,19,20,21], etc. Nevertheless, this issue remains open. The ion parameters were measured within a wide range of temperature and for different values of the *E*/*N* ratio using mass spectroscopy methods [16,17,18]. It was shown that in cryogenic discharges, at *E/N >* 10 Td, the effective temperature of the ions can exceed the temperature of the parent gas by an order of magnitude. These results are in good accordance with the conclusions of recent theoretical [19,20] and experimental [21] works. A similar result, but for dusty plasma, was obtained in [22] both experimentally and in the framework of the Monte Carlo method. It also shows that the main ions in a cryogenic discharge are atomic helium He^+^ ions if *E*/*N* exceeds 10 Td. Thus, the determination of the Debye radius and ion temperature in the cryogenic discharge remains an unsolved problem.

In this paper, the nonlinear dust-acoustic instability (with nonlinear waves excitation) in a multicomponent plasma at the buffer gas temperature of ~2 K (pressure *p* = 5 Pa, *E*/*N*~10 Td) is studied in detail. The plasma consisted of electrons, helium ions, micron-sized particles injected into the discharge, and condensed nanoparticles. As mentioned above, the properties of such plasma configuration are poorly understood. We found only one publication [2], which describes the radio-frequency RF discharge plasma containing both injected and condensed (grown) dust fractions. Ion sputtering of the surface of the injected dust particles led to the formation of a condensed fraction. Recently, such multicomponent DC discharge plasma was investigated in [23,24,25]. In [23,24], wave processes were observed; however, they were not studied in detail. The proposed research is a logical continuation of works [23,24]. Dust-acoustic waves can be an effective tool for plasma diagnostics because their main parameters can be accurately measured. In turn, the wave parameters are related to the plasma parameters by known relations. In this research, wave analysis is carried out in the framework of a simple hydrodynamic model. The screening length, ion temperature, drift velocities, and some other discharge parameters are determined independently.

## 2. Results

### 2.1. Plasma Parameters

The wave process observed in the experiment (Section 4) is rather complex, and its detailed description is beyond the scope of this work. However, our estimates allow us to reasonably describe the observed phenomena.

The process begins with an estimate of the Debye radius—*λ_D_*; this problem at ultralow temperatures was discussed in [4,9,22]. For *λ_D_*, we can write
(1)$$\lambda_{D}^{-2}=\lambda_{De}^{-2}+\lambda_{Di}^{-2}\;$$
where *λ_De_*_,*i*_ = (*ε*_0_*T_e_*_,*i*_/*e*^2^*n_e_*_,*i*_)^1/2^ are the Debye radii for electrons and ions, respectively. The width of the observed soliton-like dust density profiles is *L* ≈ 100 µm (see Section 4). As follows from theoretical work [26], large-amplitude dust-acoustic soliton widths are about 1–3 *λ_D_.* The width of the soliton was meant the width of the dust density profile. This parameter is easy to measure experimentally; thus, it is analogous to “standard candles” in astrophysics. In our case, Δ_+_ ≈ 100 μm, therefore, we have *λ_D_* ≈ 30–100 μm, which agrees well with [9,22] and significantly exceeds the estimates presented in [4,6]. We set *λ_D_* = 30 μm;that is, we assume that the width of the wave crest density profile contains at least three Debye radii (upper bound from the study by [26]). From Equation (1) and data in Table 1, it follows that *λ_D_* corresponds to the ion Debye radius *λ_D_* = *λ_Di_*, which indicates that the ion drift is subsonic. With supersonic drift, we have *λ_D_* = *λ_De_* (see, for example, [27]). In this case, by setting *T*_e_ ≈ 1 eV (Table 2), we have *λ_De_* >> *λ_Di_*; therefore, the width of the nonlinear wave profiles should be much larger than the observed value. 

Having *λ_D_* one can estimate the initial density of dust particles of the second fraction—*n*_0*d*2_. The density *n*_0*d*2_ is large enough for the effective scattering of laser radiation on the wave profile; however, in the unperturbed state, the condensed particles weakly scatter light. We assume that the particle charge is proportional to the radius as well as that *Z* = 10^3^ for *r* = 0.5 μm. Then, we obtain *Z*_2_ = 5 for *r* = *r_d_*_2_. According to [22,28,29] when the discharge temperature decreases to cryogenic values, *Z* decreases by 2–3 times, compared with normal conditions; thus, finally, we have *Z*_2_ = 1–2. For such small charges, it can be assumed that the interparticle distance is equal to the Debye radius *L*_2_ ≈ *λ_Di_* ≈ 30 μm, then *n*_0*d*2_ = *λ_Di_*^−3^ = 3.7 × 10^7^ cm^−3^. Further, from the quasi-neutrality condition *n*_0*i*_ − *n*_0*e*_ − *Z*_1_·*n*_0*d*1_ − *Z*_2_·*n*_0*d*2_ = 0, we obtain *n*_0*i*_ = *n*_0*e*_ + *Z*_1_·*n*_0*d*1_ + *Z*_2_·*n*_0*d*2_. Taking into account the data in Table 2, we can obtain *n*_0i_ = 2.3 × 10^8^ cm^−3^. Next, from (1), we obtain the estimate for the ion temperature *T_i_* = 45 K. Thus, analysis of the wave process implies *T_i_* >> *T_a_* at *E*/*N* ≈ 10 Td, which confirms the results of [16,17,18,19,20,22], obtained by other methods. The value of *n*_0*e*_ can be approximately estimated from the expression for the discharge current *I*/*S* ≈ *j* = *en*_0*e*_*u*_0*e*_, where *S* = π*R*^2^ is the cross-sectional area of the discharge tube, *j* is the current density, and *u*_0*e*_ is the drift velocity of electrons. For various glow discharge strata in different experiments *u*_0*e*_/*υ_Te_*~0.1–1, where *υ_Te_* = (*T_e_*/*m_e_*)^1/2^ is an electron thermal velocity. Then, setting *I* = 50 μA (the upper estimate for the discharge current in our experiment), we obtain *n*_0*e*_~*I*/*πR*^2^*eAυ_Te_~*10^6^–10^7^ cm^−3^, with *A = u*_0*e*_/*υ_Te_* = 0.1–1. The value of *n*_0*e*_ on the discharge axis may be several times higher than the obtained estimate due to the radial inhomogeneity of the current. Finally, we estimate the mean free path of ions by the formula *l_i_* = (*n_a_σ_in_*)^−1^ [30], where *σ_in_* is the cross section for the scattering of ions on helium atoms. In accordance with the data listed in Table IIa from [18] *σ_in_* = 10^−14^ cm^2^, with *T_a_* = 4.35 K (there are no data at *T_a_* = 2 K) and *E*/*N* = 10 Td, then *l_i_* = 5.5 μm. Thus, *λ_D_* > *l_i_* (the ions are collisional). The calculated data are provided in Table 2, according to which the forces acting on the dust particles need to be calculated.

### 2.2. The Main Forces

The ion drift velocity, in accordance with the data listed in Table IIa from [18], is *u*_0*d*2_ = 5.2·10^3^ cm/s at *E*/*N* = 10 Td, considering that the main ions are He^+^ [22]. It is worth noting that the type of ion is not so important for the estimates, since the mobilities of the He^+^, He_2_^+^, and He_3_^+^ ions are not very different [31,32]. Assuming that *T_i_* = 45 K, we can obtain an estimate for the ion thermal velocity υ*_Ti_* = (*T_i_*/*m_i_*)^1/2^ = 3.05 × 10^4^ cm/s, hence *u*_0*d*2_ << υ*_Ti_*.

In the vertical direction, the main forces to be considered are the gravitational force *F_G_*, the electric force due to a vertical discharge electric field *F_E_*, the neutral drag force *F_nd,_* and ion drag force *F_id_*. The first three forces are described by simple expressions as follows:(2)$$F_{G}=\frac{4}{3}\pi r_{d}^{3}\rho_{d}g,$$
(3)$$F_{E}=-eZ_{a}E,$$
(4)$$F_{nd}=\frac{8}{3}\sqrt{2\pi }r_{d}^{2}m_{n}n_{n}\gamma \upsilon_{Tn}\upsilon_{da}\equiv m_{d}\nu_{nd}\upsilon_{da}$$
where *r_d_* and *ρ_d_* are the radius and mass density of dust particles, *a* = 1, 2 for particles of the first and second fractions, respectively, *g* = 981 cm/s^2^, *m_n_*, *n_n_*, and $\upsilon_{Tn}=\sqrt{T_{n}/m_{n}}$ are the mass, density, and thermal velocity of the neutral gas atoms, respectively, *υ_da_* is the speed of dust particles relative to a neutral gas, *ν_nd_* is dust-neutral collision frequency, *γ* is a coefficient on the order of unity that depends on the exact processes proceeding on the particle surface, and *γ* takes values from 1 to 1 + π/8 [1].

Ion drag forces were calculated in accordance with [33] (Equation (11)). The values of the main forces are provided in Table 2 (taking into account Table 1 data).

As can be seen from Table 2, for large particles (fraction «1»), gravity is balanced by electrostatic force. Large particles are visible, so their motion parameters can be easily determined. The large particles’ speed is small, so the neutral drag force, as well as the ion drag force, can be neglected for them. Particles of fraction 2 are invisible, and only the modulation of their density caused by waves is available for observations. However, this is sufficient to determine the important parameters of particle motion. From Table 2, it follows that *F**_E_* ≈ *F**_nd_*_2_ >> *F**_G_*~*F**_id_*_2_, i.e., the electric force can be balanced only by the neutral drag force in the presence of a drift *u*_0*d*2_ ≈ 20 cm/s, directed upwards. Taking into account the data of Table 1 and Table 2 and putting *Z*_2_ = 2, we obtain *u*_0*d*2_/*C_d_*_2_ = 4.7, where $C_{d2}=\sqrt{\frac{Z_{2}^{2}n_{0d2}T_{e}T_{i}}{m_{d2\left(n_{0e}T_{i}+n_{0i}T_{e}\right)}}}$ is the dust-acoustic speed for fraction 2 [34,35]. Thus, the drift of fraction 2 is supersonic. Under these conditions, the phase state “liquid” or “gas” is most likely for fraction 2. The soliton-like wave crests move downward relative to the particles of the second fraction with a speed equal to or greater than *u*_0*d*2_. At the same time, in a fixed coordinate system, the wave speed is *V*_0_ ≤ 1 cm/s. In the first approximation, waves can be considered as standing in a fixed coordinate system because |*V*_0_/*u*_0*d*2_| << 1. A similar situation for micron size particles is described in [36]. Knowing the basic parameters of the dusty plasma, we turn to the calculation of the parameters of the nonlinear waves.

### 2.3. Hydrodynamic Wave Model

We assume that in the unperturbed state, the dust particles of the second fraction are affected only by the electric force *F_E_*_2_ and the neutral drag force *F_nd_*_2_ (due to the particle drift directed upward). Other forces are negligible (Table 2). The waves have almost no effect on the parameters of the first fraction, so we assume that *N_d_*_1_ = *n_d_*_1_/*n*_0*d*1_ = 1. The dynamics of dust particles can be described by a system of one-dimensional hydrodynamic equations. In the frame of the second fraction—namely, in the frame moving upward with the speed *u*_0*d*2_, the equations take the following form:(5)$$\frac{\partial \upsilon_{d2}}{\partial t}+\upsilon_{d2}\frac{\partial \upsilon_{d2}}{\partial z}=-\frac{eZ_{2}}{m_{d2}}\left(E+E^{\prime }\right)-\nu_{nd2}\left(\upsilon_{d2}-u_{0d2}\right)$$
(6)$$\frac{\partial n_{d2}}{\partial t}+\frac{\partial n_{d2}\upsilon_{d2}}{\partial z}=0$$
where *υ_d_*_2_, *n_d_*_2_, *m_d_*_2_ are the velocity, density, and mass of the second fraction particles, $E^{\prime }=-\partial \phi /\partial z$ is the electrostatic field of the wave, *φ* is potential, and *ν_nd_*_2_ is the dust-neutral collisions frequency, which is easy to find from Equation (4); all other parameters are listed in Table 1 and Table 2. Considering that $\sum F_{0}=0$, where *F*_0_ is zero-order forces (i.e., $\nu_{nd2}u_{0d2}-eZ_{2}E/m_{d2}=0$), Equation (5) can be rewritten as
(7)$$\frac{\partial \upsilon_{d2}}{\partial t}+\upsilon_{d2}\frac{\partial \upsilon_{d2}}{\partial z}=-\frac{eZ_{2}}{m_{d2}}E^{\prime }-\nu_{nd2}\upsilon_{d2}.$$

In the first approximation, considering $\nu_{nd2}\upsilon_{d2}=0$, in the stationary case, according to [34], Equations (6) and (7) can be converted to the following form:(8)$$N_{d2}\left(\Phi \right)=\frac{n_{d2}}{n_{0d2}}=\frac{M}{\sqrt{M^{2}+2Z_{2}\Phi }},$$
where Φ = *eφ*/*C_d_*_2_^2^*m**_d_*_2_ is the normalized potential, *M* = *V*/*C_d_*_2_ is the Mach number, and *V* is the steady-state wave velocity in the moving frame. The density of electrons and ions can be described by the Boltzmann distribution as follows:(9)$$N_{e}\left(\Phi \right)=\frac{n_{e}}{n_{0e}}=\exp\left(\frac{e\varphi }{T_{e}}\right)\equiv \exp\left(\frac{Z_{2}\Phi }{\delta_{1}+\beta \delta_{2}}\right),$$
(10)$$N_{i}\left(\Phi \right)=\frac{n_{i}}{n_{0i}}=\exp\left(-\frac{e\varphi }{T_{i}}\right)\equiv \exp\left(-\frac{Z_{2}\Phi \beta }{\delta_{1}+\beta \delta_{2}}\right),$$
where $\beta =T_{e}/T_{i}$, $\delta_{1}=n_{0e}/Z_{2}n_{0d2}$, $\delta_{2}=n_{0i}/Z_{2}n_{0d2}$. Equations (8)–(10) are connected with the stationary Poisson equation.
(11)$$\frac{d^{2}\Phi }{dS^{2}}=\frac{1}{Z_{2}}\left(\delta_{1}N_{e}-\delta_{2}N_{i}+\delta_{3}N_{d2}\right),$$
where $S=\left(z-Vt\right)/\lambda_{Di}$, $\delta_{3}=n_{0d1}/Z_{2}n_{0d2}$. The quasi-neutrality condition yields Δ_1_ − Δ_2_ + Δ_3_ + 1 = 0.

Equation (11) can have solutions in the form of a soliton, nonlinear (cnoidal) wave, or linear wave, depending on the initial conditions. To describe large-amplitude soliton-like dust-acoustic waves, we use the soliton solution of Equation (11). Such solutions of Equation (11) are well studied (see, for example, [1]). Figure 1 shows the numerically obtained soliton profiles at *M* = 1.55 and *M* = 1.7. Other parameters of the model are *β* = 222, Δ_1_ = 0.135, Δ_2_ = 3.16, Δ_3_ = 2.03 (Table 2). When integrating, the Runge–Kutta algorithm is used, the boundary conditions are as follows: Φ(−10) = −5 × 10^−4^ and Φ’(−10) = 10^−5^ at *M* = 1.7; Φ(−10) = −8·10^−4^ and Φ’(−10) = 10^−5^ at *M* = 1.55.

From Figure 1, it can be seen that the wave width is Δ*φ* ≈ 3–5 *λ_Di_*, and the width of the particle density profile is Δ*N_d2_* ≈ 1 *λ_Di_*. Our simple “cold” plasma model does not take into account the pressure of the dust fraction. However, in more realistic and complex models, the width of solitons can increase by several times (see, for example, the “warm” plasma model in [1]). Therefore, we assumed Δ*N_d2_* = 3*λ_Di_* in the above reasoning. The density of dust in the center of the soliton is increased up to 4–10 times. With such an increase in density, the soliton-like profiles can be easily observed in the optical range due to light scattering.

The electric field inside the soliton is perturbed by the value of Δ*E*’ < 0.5 V/cm (Figure 1c). Due to the fulfillment of the inequality Δ*E*’ << *E*, the nonlinear waves do not have a noticeable effect on the first fraction particles, which is confirmed experimentally. The wave velocity is *V* = 7 cm/s, which corresponds in order of magnitude to *u*_0*d*2_, but this is not enough for the observed slow wave motion downward in a fixed coordinate system. For a more accurate description of the experiment, additional research is needed. We return to this issue in the discussion.

## 3. Discussion

The reasons for the instability excitation can be ions (electrons) drift and ion drag force [37,38,39,40,41,42], dust charge perturbation [43,44,45], neutral drag force caused by the dust drift relative to a stationary neutral gas [46]. The third reason, in our opinion, dominates in the case under consideration. Indeed, first, the instability conditions obtained in [46] are satisfied, such as *n*_0*e*_/*n*_0*i*_ << 1 and *E*/*N* > *E_crit_/N*, where *E_crit_* is the minimum value of the electric field sufficient to excite the instability. Secondly, the nonlinear waves are plane, occupying almost the entire radial cross section of the tube, far beyond the boundaries of the large particles cloud. The neutral drag force can be considered constant in the radial direction. The electric forces and the ion drag force should have a radial gradient, as evinced by the shape of the large particles cloud. If the forces of *F_E_*_2_ and *F_id_*_2_ caused the instability, the solitons would have a lenticular shape. The radial homogeneity of solitons can also be explained by the small charge of condensed particles (*Z*_2_ ~ 1) since the discharge boundaries do not affect them significantly. A detailed analysis of the causes of instability is certainly of interest and will be carried out in the future.

In conclusion, we highlight the finding that continuous ion sputtering of the cone surface should lead to a loss of its mass Δ*m*, which can be estimated by the formula Δ*m* = *m_d_*_2_·*n*_0*d*2_·π·*R*^2^·*u*_0*d*2_·Δ*t*. During the experiment Δ*t* ≈ 1.2·10^3^ s, the mass loss can be neglected, as Δ*m* < 10^−6^ kg.

## 4. Experiment

The experiments were carried out in the Janis SVT-200 cryostat, which allows operation within the temperature range from 1.5 to 300 K [25]. A DC glow discharge was created in helium gas inside a vertical glass tube with an inner diameter of 2 cm. The distance between electrodes was equal to 60 cm. The experimental setup is shown schematically in Figure 2a. Polydisperse CeO_2_ particles (0.1–200 μm) were used to form a dust cloud in the lowest stratum of the discharge.

The pressure was measured using a Granville-Phillips 275 convectron, while the temperature was determined by a LakeShore 335 temperature controller connected to the TPK-1.5/60-22 semiconductor sensor calibrated within the working range of 1.5–60 K and attached to the discharge tube wall at the level of the lowest stratum. The waves were recorded using a high-speed digital camera (up to 1000 fps), a “laser knife” (with the cross section of 0.22 × 6 mm, 85 mW at 532 nm) was used for illumination of the dust cloud.

The operating temperatures were achieved by reducing the discharge current to 35 ± 15 μA at a voltage of 3.2 kV and a pressure of 5 Pa. In such a low-power mode, the temperature gradient between the anode and cathode did not exceed 2–3 K. As shown in Figure 2b, the initial shape of the dust cloud is close to spherical, the mean interparticle distance was ≈170 μm. At the same time, the particles formed the structure of the liquid type, containing randomly moving fast and slow particles. Fast particles formed intense vortex flows on the periphery of the cloud.

Approximately 20 min after the discharge had been generated, we detected scattering of laser radiation on moving local inhomogeneities as shown in Figure 3. At about the same time, the average distance between micron size particles decreased to ≈120 μm. It is clear from Figure 3, that there are no visible separate particles in the inhomogeneities. These facts indicate the appearance of a new submicron dust fraction, which was formed in the plasma during the experiment. The geometry of moving inhomogeneities indicates the excitation of dust-acoustic instability involving submicron particles. In what follows, subscripts 1 and 2 refer to injected (micron) and condensed (submicron) dust particles, respectively. As shown in [24], the appearance of fraction 2 was due to ionic sputtering of the dielectric cone material (8 on Figure 2a). The cone was used to focus the flow of electrons on the axis of the discharge tube. A flow of ions was also focused in the region of the cone and sputtered its surface. This hypothesis was confirmed during an additional experiment carried out at room temperature, which revealed an intense glow near the cone outlet. It is well known that polymer surfaces are easily sputtered in strata of the DC glow discharge [47,48]. Therefore, we can expect rather an efficient sputtering of the cone material because it contains acrylic polymer [24]. The condensed fraction properties were studied in detail in a few studies [23,24].

The shape of the condensed particles is close to spherical, and the diameter does not exceed 75 nm with a large dispersion. The appearance of the condensed fraction led to some changes in the parameters of the initial dust cloud. For example, as mentioned above, the density of large particles increased (probably due to a decrease in the charge of large particles with the appearance of an additional submicron dust fraction and, therefore, due to competition for electrons and ions). A detailed study of the effect of fraction 2 on the evolution of the cloud is beyond the scope of this work. Notably, some parameters of the discharge slowly changed during the experiment. For example, the temperature of liquid helium varied in the range from 1.6 to 2.17 K. The main parameters of the discharge are listed in Table 1.

Considering the waves in fraction 2, as can be seen from Figure 3, fraction 2 fills a wider part of the tube than fraction 1. The waves were investigated in the same axial region where the particles of fraction 1 levitated. The observed wave process was rather complicated. However, some of its features can be considered. We use the one-dimensional approximation since the waves can be considered planes. Once the axis OZ is directed vertically downward (from the anode to the cathode), it can be inferred that the wave perturbation of the submicron particle (fraction 2) density cannot be described by a harmonic function (Δn_d2_ ≠ exp(iωt + ikz)). Indeed, the width of the compression regions, Δ_+_, is noticeably smaller than the width of the rarefaction regions, Δ_−_; therefore, the waves are nonlinear. In addition, Δ_+_ ≈ 100 μm for all cases, while Δ_−_ depended on z and t. Such behavior is characteristic of strongly nonlinear waves. The amplitudes of the individual wave crests were especially large (bright bands on Figure 3). These crests had soliton-like dust density profiles. The speed of the nonlinear waves was different in different parts of the dust structure but did not exceed 1 cm/s (V ≤ 1 cm/s). According to our estimates (see below), the dust-acoustic velocity for the submicron dust fraction should significantly exceed the observed wave velocity (C_d2_ >> V). Such a low velocity of nonlinear dust-acoustic waves can be explained either within the framework of the model [49] (where the self-consistent charge of particles was taken into account), within the framework of research [50] (where dust compression waves in electrorheological dusty plasma were studied), or by the presence of submicron particle drift (second fraction particles).Under the considered experimental conditions, the drift hypothesis appears to be the most reasonable. It is important that the drift velocity of the second fraction particles should be approximately equal to the phase velocity of the waves. The effect of the observed waves on the particles of the first fraction was insignificant. Consequently, the electric field of the discharge significantly exceeded the electric field of the waves. We use standard models of a glow discharge dusty plasma for the theoretical interpretation of the experiment. The hydrodynamic model is used to calculate the parameters of the waves.

## 5. Conclusions

The nonlinear dust-acoustic instability excited in the condensed submicron fraction of dust in the four-component dusty plasma of the DC glow discharge at the gas temperature of ~2 K was studied in detail. The plasma contained injected CeO_2_ particles of micron size and condensed nanoparticles, as well as electron and ion background. The nanoscale fraction was aggregated from products of ion sputtering of the polymer cone, which performed the function of focusing the electron stream. To analyze the observed waves, a simple hydrodynamic model was used, which made it possible to estimate important discharge parameters: *λ_Di_*, *T_i_*, particle drift velocities, etc. Independently of other authors [16,17,18,19,20,21,22], it was shown that the ions in the discharge are overheated in the presence of a significant electric field. The screening length was determined by the ions (*λ_D_* ≈ *λ_Di_*). The obtained estimates for *λ_Di_* are in good agreement with the results of [9] and are several times higher than the estimates of [4,6]. This discrepancy is due to the fact that *T_i_ = T_a_* was assumed in [4,6], while our analysis resulted in *T_i_ >> T_a_*. Estimates of other discharge parameters—namely, *E*/*N*, *l_i_*, are consistent with [4].

Analysis of the acting forces showed that the second (invisible) fraction of dust drifts upward at the speed exceeding the dust-acoustic speed several times. The calculated electric field of strongly nonlinear waves proved to be an order of magnitude smaller than the field of discharge, which explains the absence of a relationship between waves and micron particles of the first fraction. The reasons for the excitation of the instability were briefly considered. The most likely cause is the neutral drag force, arising in the presence of significant drift of the second fraction particles. The linear stage of such instability was theoretically considered in [46].

In accordance with the assumptions made, the wave velocity *V* must be either equal to or slightly higher than the drift velocity *u*_0*d*2_ (i.e.,*V* ≥ *u*_0*d*2_). In our estimates, the soliton velocity and the drift velocity coincide only in order of magnitude; what is more, *V* < *u*_0*d*2_. One can achieve the fulfillment of the condition *V* = *u*_0*d*2_ in two ways. First, by increasing the velocity of the nonlinear waves to *M* ~ 4–5. Secondly, by reducing the drift velocity *u*_0*d*2_. Future research will help clarify the values of *V* and *u*_0*d*2_.

## Figures and Tables

**Figure 1 molecules-27-00227-f001:**
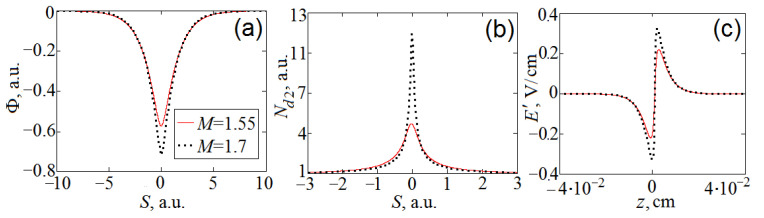
Soliton profiles with *M* = 1.55 and *M* = 1.7: (**a**) potential; (**b**) density of the second fraction particles; (**c**) electric field of the soliton.

**Figure 2 molecules-27-00227-f002:**
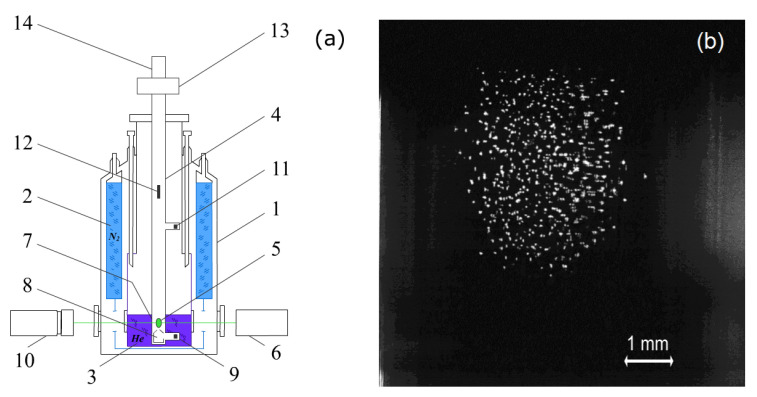
(**a**) Scheme of the experimental setup: 1—cryostat; 2—liquid nitrogen bath; 3—liquid helium bath; 4—gas discharge tube; 5—dusty plasma structure; 6—laser; 7—thermometer on the wall of the discharge tube; 8—dielectric cone; 9—cathode; 10—video camera; 11—anode; 12—dust particle injector; 13—cross-shaped connector; 14—pressure sensor; (**b**)initial dust cloud.

**Figure 3 molecules-27-00227-f003:**
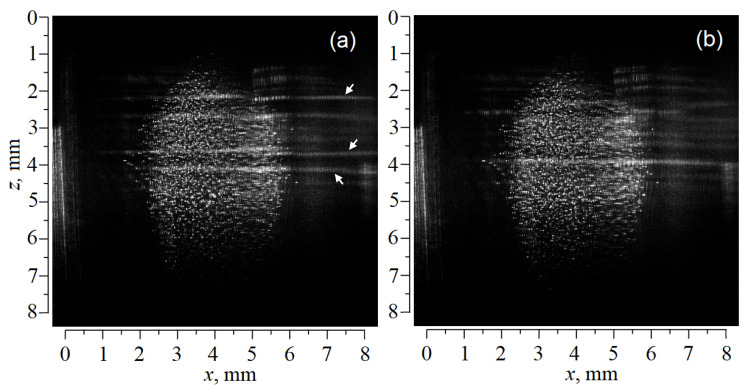
Dust cloud in the DC discharge stratum containing a mixture of the CeO_2_ particles and condensed submicron particles formed 20 min after the start of the experiment;(**a**) *t* = 0; (**b**) *t* = 30 ms. The positions of large-amplitude wave crest are indicated by arrows.

**Table 1 molecules-27-00227-t001:** The main parameters of the plasma.

Plasma Parameters	Value
Neutralgas pressure.	*P*_He_ = 5 Pa
Discharge current	*I* = 35 ± 15 μA
Discharge voltage	*U* = 3.21 kV
Neutralgas density	*n*_He_ = 1.8 × 10^23^ m^−3^
Temperature of the neutral gas (the walls of the discharge tube)	*T_a_* ≈ 2 K
Radius of the first fraction dust particles	*r_d_*_1_ ≈ 1–5 μm
Radius of the second fraction dust particles	*r_d_*_2_ ≈ 15–35 nm
Mass density of the first fraction dust particles	ρ_1_ = 7200 kg/m^3^
Mass density of the second fraction dust particles	ρ_2_ = 1100–1500 kg/m^3^
Electric field strength	*E* ≈ 2000 V/m
Reduced electric field strength	*E*/*N* ≈ 10 Td

**Table 2 molecules-27-00227-t002:** Calculated plasma parameters.

Plasma Parameters	Value
Density of the first fraction particles	*n_d_*_1_ = 3.0 × 10^5^ cm^−3^
Density of the second fraction particles	*n_d_*_2_ = 3.7 × 10^7^ cm^−3^
Charge of the first fraction dust	*Z*_1_ ≈ 500
Charge of the second fraction dust	*Z*_2_ = 2
Electron density	*n_e_* ≈ 1 × 10^7^ cm^−3^
Ion density	*n_i_* ≈ 2.3 × 10^8^ cm^−3^
Electron temperature	*T_e_* ≈ 10^4^ K
Ion temperature	*T_i_* ≈ 45 K
Electron Debye length	*λ_De_* = 2.2 × 10^−1^ cm
Ion Debye length	*λ_Di_* = 3.0 × 10^−3^ cm
Ion free path	*l_i_* = 5.5 μm
Gravitational force for the first fraction particles	*F_G_*_1_ = 1.6 × 10^−13^ N
Gravitational force for the second fraction particles	*F_G_*_2_ = 2.0 × 10^−18^ N
Electric force for the first fraction particles	*F_E_*_1_ = 1.6 × 10^−13^ N
Electric force for the second fraction particles	*F_E_*_2_ = 3.2 × 10^−16^ N
Ion drag force for the first fraction particles	*F_id_*_1_ = 3.0 × 10^−14^ N
Ion drag force for the second fraction particles	*F_id_*_2_ = 4.9 × 10^−18^ N
Neutral drag force for the first fraction particles at *u*_0*d*1_ = 5 × 10^−3^ m/s	*F_nd_*_1_ = 1.6 × 10^−15^ N
Neutral drag force for the second fraction particles at *u*_0*d*2_ = 0.2 m/s, γ = 1 + π/8	*F_nd_*_2_ = 2.2 × 10^−16^ N
Electron thermal velocity	*υ_Te_* = 4 × 10^7^ cm/s
Ion thermal velocity	*υ_Ti_* = 3 × 10^4^ cm/s

## Data Availability

The data presented in this study are available on request from the authors.

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
