# Peer review of "Dust-Acoustic Nonlinear Waves in a Nanoparticle Fraction of Ultracold (2K) Multicomponent Dusty Plasma"

_molecules, 2021, doi:10.3390/molecules27010227_

Round 1

Reviewer 1 Report

Dear authors,

The manuscript molecules-1474080-peer-review-v2, entitled ‘Dust-acoustic nonlinear waves in a nanoparticle fraction of ultracold (2K) multicomponent dusty plasma’ presents the nonlinear dust-acoustic instability in a multicomponent plasma at the buffer gas temperature of ~ 2 K. Wave analysis is carried out in the framework of a simple hydrodynamic model. The screening length, ion temperature, drift velocities, and some other discharge parameters are determined independently.

Their conclusion is that the calculated electric field of strongly nonlinear waves turned out to be an order of magnitude smaller than the field of discharge, which explains the absence of a relationship between waves and micron particles of the first fraction. The reasons for excitation of the instability are briefly considered.

Computational details are provided by the authors separately in each section. The methodology used by the authors is well presented, clearly and in detail, on every step.

The manuscript is neatly written. Graphics are of good quality, as in the requirements of the journal, and express the values determined from the numerical calculation.

The conclusions are in brief but sustained by the findings of numerical / computational modeling.

Author Response

We thank the referee for the work done.

Reviewer 2 Report

In this manuscript the authors report an interesting and unique experiment on nonlinear waves in multicomponent ultracold (2K) dusty plasma. The theoretical interpretation can probably be questioned, but the experimental results deserve to be published. The manuscript is reasonably well written, presentation of results seem appropriate. I would recommend publication in Molecules, after the authors consider several points that can be improved.

  1. On p. 1, line 39-41 the authors mention that the screening radius in dusty plasma can considerably exceed the ion Debye radius in plasma. One possible reason for this can be non-linear screening of a particle charge as discussed for instance by Hutchinson [PHYSICS OF PLASMAS 20, 083701 (2013)] and Semenov [PHYSICS OF PLASMAS 22, 053704 (2015)]. This can be mentioned.
  2. On p. 4, lines 139-142 the authors state that the actual wave velocity they observed in experiment was smaller than the dust-acoustic velocity they estimated, if I get this correctly. Similar observation has been recently reported by Schwabe [New J. Phys. 22 (2020) 083079] and interpreted in terms of additional attractive interactions caused by ion wake-related effects. It would be good to comment on this.
  3. On p. 5, line 200 there seems to be a misprint: ion thermal velocity should be 3\times 10^4 cm/s, not 3\times 10^2 (see also Table 2). Correct or make consistent.
  4. In section 4 a possible reason for the observed instability is discussed. In a recent paper by Khrapak and Yaroshenko [Plasma Phys. Control. Fusion 62, 105006 (2020)] the ion drift instability mechanism in a strongly coupled highly collisional dusty plasma has been discussed in detail, although employing linear analysis. This can probably be relevant.
  5. In Sec. 4 and 5 the authors repeatedly state that the instability is likely caused by the neutral drag force. This is misleading in my opinion. The neutral drag force introduces dissipation and thus it can only stabilize instabilities of various kind, but not cause instability. Please, reformulate this.       
